# Simple data balancing achieves competitive worst-group-accuracy

**Badr Youbi Idrissi**[*]                                      BADRYOUBIIDRISSI@GMAIL.COM
*CentraleSupélec, Université Paris-Saclay, France*

**Martín Arjovsky**                                      MARTINARJOVSKY@GMAIL.COM
*INRIA - PSL, Paris, France*

**Mohammad Pezeshki**[†]                              MOHAMMAD.PEZESHKI@UMONTREAL.CA
*Mila, Université de Montréal, Canada*

**David Lopez-Paz**                                                  DLP@FB.COM
*Facebook AI Research, Paris, France*

**Editors:** Bernhard Schölkopf, Caroline Uhler and Kun Zhang

## Abstract

We study the problem of learning classifiers that perform well across (known or unknown) groups of data. After observing that common worst-group-accuracy datasets suffer from substantial imbalances, we set out to compare state-of-the-art methods to simple balancing of classes and groups by either subsampling or reweighting data. Our results show that these data balancing baselines achieve state-of-the-art-accuracy, while being faster to train and requiring no additional hyper-parameters. In addition, we highlight that access to group information is most critical for model selection purposes, and not so much during training. All in all, our findings beg closer examination of benchmarks and methods for research in worst-group-accuracy optimization.

## 1. Introduction

Machine learning classifiers achieve excellent test average classification accuracy when both training and testing data originate from the same distribution (Vapnik, 1995; LeCun et al., 2015). In contrast, small discrepancies between training and testing distributions cause these classifiers to fail in spectacular ways (Alcorn et al., 2019). While training and testing distributions can differ in multiple ways, we focus on the problem of *worst-group-accuracy* (Sagawa et al., 2019). In this setup, we discriminate between multiple *classes*, where each example also exhibits some (labeled or unlabeled) *attributes*. We call each class-attribute combination a *group*, and assume that the training and testing distributions differ in their group proportions. Then, our goal is to learn classifiers maximizing worst test performance across groups.

Optimizing worst-group-accuracy is relevant because it reduces the reliance of machine learning classifiers on *spurious correlations* (Arjovsky et al., 2019), that is, patterns that discriminate classes only between specific groups (Shah et al., 2020; Geirhos et al., 2018, 2020). The problem of worst-group-accuracy is also related to building fair machine learning classifiers (Barocas et al., 2019), where groups may have societal importance (Datta et al., 2014; Chouldechova, 2017; Rahmattalabi et al., 2020; Metz and Satariano, 2020).

Maximizing worst-group-accuracy is an active area of research, producing two main strands of methods (reviewed in Section 3). On the one hand, there are methods that consider access to

---

[*] Work done while interning at Facebook AI Research and INRIA, Paris, France
[†] Work done while interning at Facebook AI Research, Montréal, Canada

attribute information during training, such as the popular group Distributionally Robust Optimization (Sagawa et al., 2019, gDRO). On the other hand, there are methods that consider access only to class information during training, such as the recently proposed Just Train Twice (Liu et al., 2021, JTT). Unsurprisingly, methods using attribute information achieve the best worst-group-accuracy. But, since labeling attributes for all examples is a costly human endeavour, alternatives such as JTT are of special interest when building machine learning systems featuring strong generalization and requiring weak supervision.

This work takes a step back and studies the characteristics of four common datasets to benchmark worst-group-accuracy models (CelebA, Waterbirds, MultiNLI, CivilComments). In particular, we observe that these datasets exhibit a large class imbalance which, in turn, correlates with a large group imbalance (Section 2). In light of this observation, we study the efficacy of training systems under data subsampling or reweighting to balance classes and groups (Section 4).

Our experiments (Section 5) provide the following takeaways:

- Due to class or attribute imbalance, simple data balancing baselines achieve competitive performance in four common worst-group-accuracy benchmarks, are faster to train, and require no additional hyper-parameters.

- While we obtained the best results by balancing groups, simple class balancing is also a powerful baseline when attribute information is unavailable.

- Access to attribute information is most critical for model selection (in the validation set), and not so much during training.

- We recommend practitioners to try data subsampling first, since (i) it is faster to train, (ii) is less sensitive to regularization hyper-parameters, and (iii) has a stable performance during long training sessions.

- Given the efficacy of subsampling methods, question the mantra "just collect more data", particularly when optimizing for test *worst-group-accuracy* (as opposed to the classic goal of optimizing test *average* accuracy).

In essence, we beg for closer examination of both benchmarks and methods for future research in worst-group-accuracy optimization.

## 2. Popular worst-group-accuracy benchmarks

We consider datasets $\{(x_i, y_i, a_i)\}_{i=1}^n$, where each example is a triplet containing an input $x_i$, a class label $y_i$, and an attribute label $a_i$. The sequel studies four popular worst-group accuracy benchmarks that follow this structure.

- **CelebA** (Liu et al., 2015; Sagawa et al., 2019) consists of images of aligned celebrity faces. Each face image is annotated with multiple traits. Here, our task is to classify if the person has blond hair. The attribute indicates whether the person in the image is male or female.

| Dataset | Target | Group Counts | | Class Counts | $\hat{\mathbf{P}}(\mathbf{Y}=\mathbf{y}|\mathbf{A}=\mathbf{a})$ | | $\hat{\mathbf{P}}(\mathbf{Y}=\mathbf{y})$ |
|---|---|---|---|---|---|---|---|
| | $\downarrow y \quad a \rightarrow$ | Female | Male | | Female | Male | |
| CelebA | Blond | 22880 | 1387 | 24267 | 24.2% | 2.0% | 14.9% |
| | Not blond | 71629 | 66874 | 138503 | 75.8% | 98.0% | 85.1% |
| | | Water | Land | | Water | Land | |
| Waterbirds | Land bird | 56 | 1057 | 1113 | 1.6% | 85.2% | 23.2% |
| | Water bird | 3498 | 184 | 3682 | 98.4% | 14.8% | 76.8% |
| | | Identity | Other | | Identity | Other | |
| CivilComments | Non toxic | 90337 | 148186 | 238523 | 83.6% | 92.1% | 88.7% |
| (Coarse) | Toxic | 17784 | 12731 | 30515 | 16.4% | 7.9% | 11.3% |
| | | No negation | Negation | | No negation | Negation | |
| | Contradiction | 57498 | 11158 | 68656 | 30.0% | 76.1% | 33.3% |
| MultiNLI | Entailment | 67376 | 1521 | 68897 | 35.2% | 10.4% | 33.4% |
| | Neutral | 66630 | 1992 | 68622 | 34.8% | 13.6% | 33.3% |

Table 1: Class and group counts for four popular worst-group-accuracy benchmarks. These datasets exhibit large class ($y$) and group imbalance. In particular, class probabilities shift significantly when conditioning on the attribute ($a$) value. For instance, the CelebA dataset has only $15\%$ of examples of class "blond". Moreover, the probability of "blond" is different when the attribute value is "female" ($24\%$) or "male" ($2\%$), creating a spurious correlation.

- **Waterbirds** (Wah et al., 2011; Sagawa et al., 2019) contains images of birds cut and pasted on different backgrounds. The task is to classify specimens into water birds or land birds. The attribute indicates whether the bird appears on its natural habitat or not.

- **MultiNLI** (Williams et al., 2017; Sagawa et al., 2019) is a dataset containing pairs of sentences. The task is to classify the relationship between the two sentences as being a contradiction, an entailment, or none of the two. The attribute indicates the presence of the negation words: 'no', 'never', 'nobody' or 'nothing' in the second sentence. The presence of these words makes the 'contradiction' label more likely in this dataset. We've kept this limited list of negation words to have comparable results with previous literature.

- **CivilComments** (Borkan et al., 2019; Koh et al., 2021) is a dataset containing comments from online forums. The task is to classify whether a comment is toxic or not. There are multiple attributes annotating the content of each comment, relating to: male, female, LGBT, black, white, Christian, Muslim, other religion. Following Sagawa et al. (2019), we consider a *coarse* version of the CivilComments dataset to train gDRO. This coarse version provides a binary attribute indicating if any of the eight attributes listed above appears in the comment.

Table 1 lists the number of examples per class and group for these four datasets. The data reveals that three out of four datasets exhibit a large class imbalance, and that all of them exhibit a large group imbalance. Furthermore, these imbalances are highly correlated: class probabilities vary significantly when conditioning on the attribute value. In the Waterbirds dataset, class probabilities invert when swapping attribute values. In the MultiNLI dataset, the class "contradiction" is much more likely

when there is a negation in the second sentence. In CelebA, it is unlikely to find examples from the "male" class when the attribute is "blond". Therefore, these datasets contain spurious correlations helpful to discriminate only between some groups. When such groups represent most of the dataset, learning algorithms latch onto the spurious correlations, and resort to memorization to achieve zero training error (Sagawa et al., 2020).

These observations immediately motivate training classifiers under data subsampling or reweighting to balance out classes and groups. After group balancing, we expect the spurious correlations between classes and attributes to vanish, improving test worst-group-accuracy. Before exploring the efficacy of these simple balancing baselines, we first review some popular state-of-the-art methods proposed to optimize worst-group-accuracy.

## 3. Popular worst-group-accuracy methods

We review three popular methods appearing in the literature of worst-group-accuracy optimization.

- **Empirical Risk Minimization** (Vapnik, 1995, **ERM**) chooses the predictor minimizing the empirical risk $\frac{1}{n} \sum_{i=1}^{n} \ell(f(x_i), y_i)$. ERM does not use attribute labels.

- **Just Train Twice** (Liu et al., 2021, **JTT**) proceeds in two steps. First, JTT trains an ERM model for a small amount of epochs $T$. Assuming that this "simplistic" ERM model classifies examples based on spurious correlations, its errors should correlate to the subset of examples where the spurious pattern does not appear. Following this assumption, JTT trains a final ERM model on a dataset where the mistakes from the "simplistic" ERM model appear $\lambda_{\text{up}}$ times. JTT does not use attribute labels.

- **Group Distributionally Robust Optimization** (Sagawa et al., 2019, **gDRO**) minimizes the maximum loss across groups: $\sup_{q \in \Delta_{|G|}} \sum_{g=1}^{|G|} \frac{q_g}{n_g} \sum_{i=1}^{n_g} \ell(f(x_i), y_i)$, where $G = Y \times A$ is the set of all groups, $\Delta_{|G|}$ is the $|G|-$dimensional simplex and $n_g$ is the number of examples from group $g \in G$ contained in the dataset. Therefore, gDRO uses attribute labels. In particular, gDRO allocates a dynamic weight $q_g$ to the minimization of the empirical loss of each group, proportional to its current error.

**Other methods** The literature in robust optimization is flourishing, so the comparison of all possible methods renders itself impossible. Some further examples of robust learners not using attribute information are Learning from Failure (Nam et al., 2020), the Too-Good-to-be-True prior (Dagaev et al., 2021), Spectral Decoupling (Pezeshki et al., 2020), Environment Inference for Invariant Learning (Creager et al., 2021), and the GEORGE clustering algorithm (Sohoni et al., 2020). Other examples of methods that use attribute information include Conditional Value at Risk (Duchi et al., 2019), Predict then Interpolate (Bao et al., 2021), Invariant Risk Minimization (Arjovsky et al., 2019), and a plethora of domain-generalization algorithms (Gulrajani and Lopez-Paz, 2020).

## 4. Simple data balancing baselines

Given the class and group imbalance shown in Table 1, we explore the effectiveness of four data balancing baselines on worst-group-accuracy:

- Subsampling large classes (**SUBY**), so every class is the same size as the smallest class. Such subsampling is performed once and fixed before training starts. This baseline *does not use* attribute labels.

- Similarly, subsampling large groups (Sagawa et al., 2020, **SUBG**), so every group is the same size as the smallest group. This baseline *does use* attribute labels.

- Reweighting the sampling probability of each example, so mini-batches are class-balanced in expectation (**RWY**). This baseline *does not use* attribute labels.

- Similarly, reweighting the sampling probability of each example, so mini-batches are group-balanced in expectation (**RWG**). This baseline *does use* attribute labels.

**Toy example**  To motivate our baselines, we consider a synthetic logistic regression example (Sagawa et al., 2020, Section 5.1.). The classes $y \in \{-1, +1\}$ are dependent on two attributes $a_{core}, a_{spu} \in \{-1, +1\}$ with correlations $\rho_{core}, \rho_{spu} \in [-1, 1]$. While $\rho_{core}$ remains invariant between the training and the test data, $\rho_{spu}$ varies from the training to the test set. Each attribute dictates a Gaussian distribution over input features. In particular, each input example $x$ is a concatenation of the following three components,

$$
x := \begin{bmatrix} \gamma_{\text{spu}} & x_{\text{spu}} \\ \gamma_{\text{core}} & x_{\text{core}} \\ \gamma_{\text{noise}} & x_{\text{noise}} \end{bmatrix} \in \mathbb{R}^{d+2}, \qquad \text{where} \qquad \begin{aligned} \mathbb{P}(x_{spu} \mid y) &= \mathcal{N}(a_{spu}, \sigma^2) \in \mathbb{R}, \\ \mathbb{P}(x_{core} \mid y) &= \mathcal{N}(a_{core}, \sigma^2) \in \mathbb{R}, \\ \mathbb{P}(x_{noise} \mid y) &= \mathcal{N}(0, \sigma^2) \in \mathbb{R}^d, \end{aligned} \qquad (1)
$$

where $(\sigma^2, \gamma_i)$ are the variance and scale of each of the features. The scaling factors $\gamma_i$ control the rate at which the model learns each feature: the larger $\gamma_i$, the faster the model learns about $x_i$. Moreover, the noise features $x_{\text{noise}}$ are independent from the class labels $y$, and therefore uncorrelated *in expectation*. However, in over-parameterized settings where $d$ is greater than the number of training examples, there exists an *empirical* correlation between the noise features and the class labels. Therefore, and depending on the values of $\gamma_i$, over-parametrized models can exploit noise features to memorize training examples on their path to achieving zero training error.

Figure 1 implements one instance of this example where $\rho_{\text{core}} = 1$ and $\rho_{\text{spu}} = 0.8$. Furthermore, $\gamma_{\text{spu}} = 4$, $\gamma_{\text{core}} = 1$, $\gamma_{\text{noise}} = 20$, and $\sigma = 0.15$. Given these correlation and scaling coefficients, the spurious feature is learnable much faster than the core and noise features. As shown on the first two panels of Figure 1, a vanilla ERM model mainly relies on spurious features, and therefore achieves poor test worst-group-accuracy. On the other hand, subsampling the majority group (SUBG) decorrelates the spurious feature from the labels, leading to a model that relies on the core feature, discards the spurious feature, and achieves good test worst-group-accuracy. While reweighting groups (RWG) also solves this toy example, one has to pay special attention to model selection, since test worst-group-accuracy degrades as the number of training iterations increases. We note that the probability of misclasifying when using just $x_{\text{core}}$ is $P(yx_{\text{core}} < 0) = P(y(1 + y\sigma Z) < 0) = P\left(Z < -\frac{y}{\sigma}\right) = 1 - \Phi\left(-\frac{y}{\sigma}\right) = 1 - \Phi\left(\frac{1}{\sigma}\right)$ (since $\Phi$ is symmetrical) which is $10^{-11}$ in this particular case. This means that the problem is separable using just $x_{\text{core}}$ with high probability.

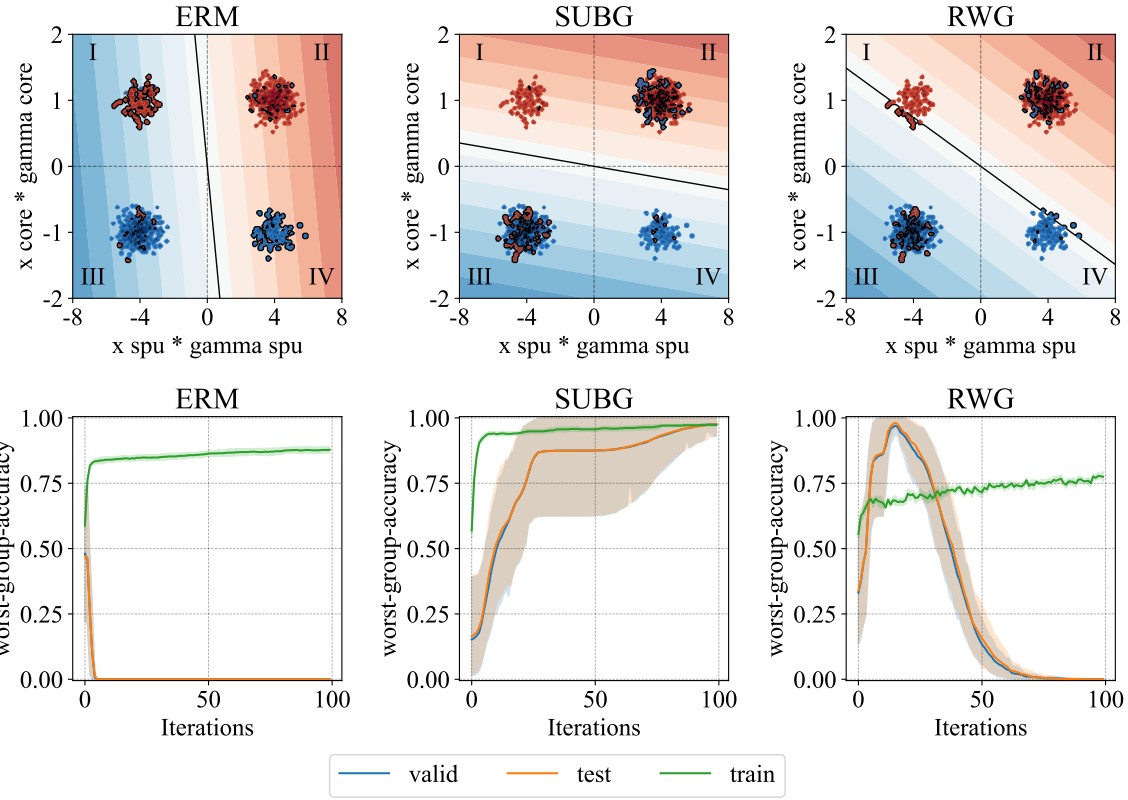

Figure 1: A linear binary classification task with a spurious feature ($x$-axis, ranging from -8 to 8), a core feature ($y$-axis, ranging from -2 to 2), and 1200 noise features (not depicted, Normally distributed). Each class contains a majority group (quadrants II and III) and a minority group (quadrants I and IV). Shades of red show the predicted probability of class +1 and shades of blue the predicted probability of class -1, under the classifier $w$. The value of each position $x$ on the heatmap is sigmoid($w^T \tilde{x}$), where we get $\tilde{x}$ by adding the noise vector of the sample in the training set closest to $x$. This enables us to visualize the regions of the 2D space ($x_{\text{spu}}, x_{\text{core}}$) where the model uses the noise vectors to predict. We depict the performance of three models at their best iteration with respect to validation worst-group-accuracy. **On the left**, an ERM model finds the easy solution of using (i) the spurious feature to discriminate between majority examples of each class, and (ii) the noise features to memorize the minority examples of each class (shown as small neighbourhoods). This leads to poor test worst-group-accuracy. **On the middle**, subsampling the majority groups (SUBG) of each class decorrelates the spurious feature and the class label, guiding the model to rely on the core feature. This leads to improved test worst-group-accuracy. **On the right**, balancing groups by data reweighting (RWG) also achieves good test worst-group-accuracy, but only when early-stopping the training process carefully. The figures are averages over eight random seeds.

## 5. Experiments

We implement ERM, JTT, gDRO, SUBY, SUBG, RWY and RWG, as well as the necessary infrastructure to experiment on the Waterbirds, CelebA, MultiNLI, and CivilComments benchmarks. Our implementation follows closely the ones of (Sagawa et al., 2019, gDRO) and (Liu et al., 2021, JTT). For the image datasets Waterbirds and CelebA, we train ResNet50 models pre-trained on ImageNET (He et al., 2016) using the SGD optimizer. For the NLP datasets MultiNLI and CivilComments, we train BERT models pre-trained on Book Corpus and English Wikipedia (Devlin et al., 2018) using the AdamW optimizer (Loshchilov and Hutter, 2017). We tune the learning rate in $\{10^{-5}, 10^{-4}, 10^{-3}\}$, weight decay in $\{10^{-4}, 10^{-3}, 10^{-2}, 10^{-1}, 1\}$, and JTT's $\lambda_{up}$ in $\{4, 5, 6, 20, 50, 100\}$. We tune the batch size in $\{2, 4, 8, 16, 32, 64, 128\}$ for CelebA and Waterbirds and $\{2, 4, 8, 16, 32\}$ for MultiNLI and CivilComments. We tune JTT's $T$ in $\{40, 50, 60\}$ for Waterbirds, $\{1, 5, 10\}$ for CelebA, and $\{1, 2\}$ for MultiNLI and CivilComments. We fix gDRO's $\eta$ to 0.1 We allow 50 random combinations of hyper-parameters for each method and dataset. In contrast to previous literature, we run each hyper-parameter random combination 5 times to compute the average and standard deviation of the reported test worst-group-accuracies. These error-bars relate to data shuffling, data subsampling, and random initialization of last linear layers. We train Waterbirds for 360 epochs, CelebA for 60 epochs, and both MultiNLI and CivilComments for 7 epochs. We select best models (hyper-parameter combination and epoch) by computing the worst-group-accuracy on a validation set. The table of the best hyper-parameters is in table 5 in the appendix. Our code is available at https://github.com/facebookresearch/BalancingGroups.

### 5.1. Results

Table 2 reports test worst-group-accuracies for all methods and benchmarks. While some methods do not require the use of attribute labels for training, we emphasize that *all methods require a validation set with attribute labels* to perform model selection. As shown in the second row of Table 3, the performance of all methods degrades when one performs model selection based on the average validation accuracy (e.g., not assuming access to attribute labels in the validation set). Table 2 also lists the number of hyper-parameters tuned by each method, four being the minimal achieved by ERM, SUBG, SUBY, RWG, RWY (learning rate, weight decay, batch size, early stopping epoch). In summary, reweighting baselines perform competitively: SUBG scores only 1.7 points less than gDRO on average, while RWY scores 2.1 points more than JTT on average. Subsampling SUBY performs below its reweighting counterpart RWY, while SUBG outperforms RWG by a small margin. Finally, the fourth row of Table 3 reports the running times employed to find the best models discussed above. This shows that ERM and RWY is 1.9 times faster than its competitor JTT, and that RWG is 1.2 times faster than its competitor gDRO. The subsampling baselines are 3.8 times faster than JTT and 7 times faster than gDRO while only having slightly worse worst-group-accuracy.

### 5.2. Analysis of exceptions

Table 2 shows two exceptions to our claim that balancing baselines have competitive results with more complicated methods. The performance of gDRO largely surpasses the rest on MultiNLI with a 8.4 points difference with the second best method. We conjecture that gDRO is performing best in Multi-NLI because of the nature of the dataset. Indeed, the spurious attribute "presence of: 'no', 'never', 'nobody' or 'nothing' in the second sentence" only helps the classifier in a small proportion

| Method | #HP | Groups | Worst Acc | | | | Average |
|--------|-----|--------|-----------|---|---|---|---------|
| | | | CelebA | Waterbirds | MultiNLI | CivilComments | |
| ERM | 4 | No | 79.7±3.7 | 85.5±1.0 | 67.6±1.2 | 61.3±2.0 | 73.5 |
| JTT | 6 | No | 75.6±7.7 | 85.6±0.2 | 67.5±1.9 | 67.8±1.6 | 74.1 |
| RWY | 4 | No | 82.9±2.2 | 86.1±0.7 | 68.0±1.9 | 67.5±0.6 | 76.2 |
| SUBY | 4 | No | 79.9±3.3 | 82.4±1.7 | 64.9±1.4 | 51.2±3.0 | 69.6 |
| RWG | 4 | Yes | 84.3±1.8 | 87.6±1.6 | 69.6±1.0 | 72.0±1.9 | 78.4 |
| SUBG | 4 | Yes | 85.6±2.3 | 89.1±1.1 | 68.9±0.8 | 71.8±1.4 | 78.8 |
| gDRO | 5 | Yes | 86.9±1.1 | 87.1±3.4 | 78.0±0.7 | 69.9±1.2 | 80.5 |

Table 2: Averages and standard deviations of test worst-group-accuracies for all methods and datasets. #HP is the number of tuned hyper-parameters. Simple data balancing baselines match the performance of state-of-the-art methods within error bars, with two exceptions. Green backgrounds indicate datasets where algorithms exhibit a statistically different performance at a significance level of $\alpha = 0.05$. This is determined using an Alexander-Govern test for the equality of means of multiple sets of samples with heterogeneous variance (Alexander and Govern, 1994). All algorithms not using attribute information perform similarly with the exception of SUBY, under-performing in CivilComments. All algorithms using attribute information perform similarly, with the exception of gDRO being better on MultiNLI.

| | ERM | JTT | RWY | SUBY | RWG | SUBG | gDRO |
|--------|-----|-----|-----|------|-----|------|------|
| Best test worst-group-accuracy | 73.5 | 74.1 | 76.2 | 69.6 | 78.4 | 78.8 | 80.5 |
| ↳ w/o attributes in validation set | -15.6 | -19.7 | -13.1 | -24.4 | -17.2 | -10.4 | -14.9 |
| ↳ w/o regularization | -9.4 | -9.9 | -6.5 | -8.5 | -12 | -1.5 | -10.3 |
| Minutes per epoch | 39 | 74 | 39 | 19 | 33 | 5 | 39 |

Table 3: Some ablations on the experimental results, averaged over datasets. The **first row** shows the best results test worst-group-accuracy, averaged across datasets, obtained by employing a validation set with attribute annotations and allowing model regularization. The **second row** shows the drop in test worst-group-accuracy when performing model selection based on *average* validation accuracy (no attribute annotations). The **third row** shows the drop in test worst-group-accuracy when performing model selection only amongst those models with weak regularization (no early stopping, weight-decay $10^{-4}$). The **fourth row** shows median running time per epoch (in minutes). SUBG is the only algorithm whose performance does not degrade when taking out regularization.

of the data (see table 1). It is therefore not a dominating spurious correlation and the classifier still needs to extract other features to achieve good accuracy. Group dro minimizes a soft maximum of the group losses, thus enabling a more flexible way to penalize this mild spurious correlation. Therefore, the simpler baselines are less effective in this case because balancing either through subsampling or reweighting is a strong measure that is meant to completely decorrelate the spurious feature with the label. The gains of getting rid of this mild spurious correlation are canceled by the harsh capacity control imposed by balancing. Indeed, subsampling throws away a big proportion of the data, and reweighting has a similar effect when stopped early, which is the case for MultiNLI (training for only a few epochs). We provide an illustrative example of the above in figure 3 in the appendix.

SUBY, on the other hand, seems to consistently underperform on the datasets in this paper, even performing worse than ERM. We therefore don't recommend its use in practice. One possible explanation for its poor performance is the fact that worst group samples could easily not be picked at all during subsampling, because of their small number, and lead to poor performance on those groups.

From these exceptions, we conclude that simple balancing baselines might only work on simpler cases where the spurious correlation is present in a big majority of the data. In more nuanced cases, such as MultiNLI, where the spurious correlation doesn't dominate the data, more complex methods such as gDRO might be useful.

### 5.3. Hyper-parameter analysis

Table 4 summarizes the top 5 best hyper-parameters for each dataset and method, together with their associated test worst-group-accuracies. We make three observations. First, the range of test worst-group-accuracies (top 1st worst-group-accuracy - top 5th worst-group-accuracy) is smaller for methods accessing group information. This means that if we used less than 50 hyperparameter tuning runs, we would still get a good worst-group-accuracy, which implies that these methods are less sensitive to hyper-parameter choice. Second, methods are most sensitive to the choice of learning rate, with multiple sets of top-5 runs preferring the same value. Third, RWG prefers small-capacity models by choosing small learning rates, high weight decays, and early epochs.

### 5.4. Evolution of worst-group-accuracy during training

Figure 2 shows the evolution of the train and test worst-group-accuracy for all methods and datasets. First, we observe that RWY, RWG, and gDRO peak in worst-group-accuracy early, and then degrade in performance. On the contrary, SUBG has a more stable performance during long sessions of training, for all datasets and especially in Waterbirds. Second, there is a consistent generalization gap for all methods and datasets regardless of regularization strength. Since some models reach $100\%$ *train* worst-group-accuracy, they must have memorized some of the worst-group examples.

### 5.5. Differences between reweighting and subsampling groups

While similar at a first glance, training models with data subsampling or reweighting may lead to different decision boundaries due to different interactions with regularization (Sagawa et al., 2020). For instance, in the absence of regularization, logistic regression converges to the maximum margin classifier (Soudry et al., 2018) in linear realizable problems. Therefore, since any strictly positive reweighting of example probabilities *has no effect on support vectors*, we conclude that regularization

| Dataset | Groups | Method | Hyperparameters | | | | Worst Acc | |
|---|---|---|---|---|---|---|---|---|
| | | | $\log_{10}(\text{LR})$ | $\log_{10}(\text{WD})$ | Epoch | Batch Size | Range | Δ |
| CelebA | No | ERM | -3.4±0.5 | -1.0±0.0 | 37.1±5.0 | 128.0±0.0 | [75.4, 80.8] | 5.4 |
| | | JTT | -3.4±0.9 | -1.8±0.4 | 30.8±6.4 | 48.0±50.3 | [70.6, 76.3] | 5.8 |
| | | RWY | -4.6±0.5 | -1.4±0.5 | 14.0±8.7 | 2.8±1.1 | [78.9, 82.9] | 4.1 |
| | | SUBY | -4.4±0.9 | -1.2±0.4 | 31.4±14.4 | 42.0±54.3 | [78.4, 79.9] | 1.4 |
| | Yes | RWG | -5.0±0.0 | -1.0±0.0 | 6.0±4.1 | 36.8±26.3 | [82.8, 84.4] | 1.7 |
| | | SUBG | -4.2±0.4 | -2.0±1.2 | 27.0±11.5 | 4.8±1.8 | [83.9, 86.6] | 2.7 |
| | | gDRO | -5.0±0.0 | -3.2±1.3 | 15.4±0.7 | 64.0±0.0 | [86.7, 87.4] | 0.8 |
| Civil Comments | No | ERM | -3.8±0.4 | -3.6±0.5 | 3.6±0.9 | 6.8±5.6 | [60.4, 61.3] | 0.9 |
| | | JTT | -5.0±0.0 | -1.8±1.5 | 4.5±0.4 | 25.6±8.8 | [62.6, 68.3] | 5.7 |
| | | RWY | -3.6±0.5 | -3.6±0.5 | 4.3±0.8 | 10.8±12.1 | [52.6, 68.3] | 15.7 |
| | | SUBY | -3.4±0.9 | -3.4±0.9 | 3.1±0.6 | 25.6±8.8 | [49.2, 51.2] | 2.1 |
| | Yes | RWG | -4.8±0.4 | -2.4±0.5 | 2.5±0.4 | 6.4±5.4 | [71.4, 72.0] | 0.6 |
| | | SUBG | -3.2±0.4 | -4.0±0.0 | 3.5±0.5 | 17.6±8.8 | [70.2, 71.8] | 1.6 |
| | | gDRO | -3.2±0.4 | -3.4±0.5 | 3.5±1.6 | 26.4±12.5 | [68.0, 69.9] | 1.9 |
| MultiNLI | No | ERM | -3.8±0.4 | -4.0±0.0 | 4.6±0.4 | 8.4±13.2 | [65.6, 67.6] | 2.0 |
| | | JTT | -5.0±0.0 | -2.2±0.8 | 4.3±0.9 | 4.4±2.2 | [65.3, 67.5] | 2.1 |
| | | RWY | -3.8±0.4 | -3.8±0.4 | 4.1±0.9 | 9.2±6.6 | [60.0, 68.0] | 8.0 |
| | | SUBY | -3.8±0.8 | -3.0±0.0 | 3.8±0.4 | 9.2±6.6 | [56.2, 64.9] | 8.7 |
| | Yes | RWG | -4.8±0.4 | -2.4±0.5 | 2.2±0.4 | 9.2±12.8 | [68.1, 69.8] | 1.7 |
| | | SUBG | -3.4±0.5 | -3.2±0.8 | 5.3±0.3 | 13.6±12.4 | [68.5, 68.9] | 0.4 |
| | | gDRO | -4.0±0.0 | -3.4±0.5 | 5.1±0.8 | 19.2±12.1 | [76.4, 78.0] | 1.5 |
| Waterbirds | No | ERM | -4.2±0.4 | -2.8±1.3 | 207.6±107.5 | 4.0±2.4 | [79.4, 85.6] | 6.2 |
| | | JTT | -3.6±0.5 | -2.8±1.1 | 187.9±117.6 | 3.6±0.9 | [83.7, 85.6] | 2.0 |
| | | RWY | -4.4±0.5 | -2.2±0.8 | 129.6±84.4 | 4.0±2.4 | [83.2, 86.1] | 2.9 |
| | | SUBY | -4.8±0.4 | -3.4±0.9 | 238.6±47.6 | 2.0±0.0 | [79.2, 82.4] | 3.2 |
| | Yes | RWG | -5.0±0.0 | -1.0±1.0 | 88.6±100.9 | 36.0±52.9 | [85.4, 87.6] | 2.2 |
| | | SUBG | -4.0±0.0 | -2.6±1.1 | 192.0±44.7 | 4.8±1.8 | [87.9, 89.1] | 1.1 |
| | | gDRO | -5.0±0.0 | -0.6±0.5 | 23.4±27.2 | 4.0±2.4 | [86.4, 88.2] | 1.8 |

Table 4: Means and standard deviations of the hyper-parameters chosen by the top 5 runs for each dataset and method. The last column shows the range of the associated test worst-group-accuracies. Blue indicates low values, yellow indicates large values.

is necessary for reweighting to have any effect on the resulting classifier. In contrast, subsampling changes the support of the dataset, likely removing support vectors and affecting the final classifier *even in the absence of regularization.*

While the previous are proven facts only for linear problems, Table 3 shows similar findings for the over-parametrized deep models used in this work. In particular, the reweighting methods

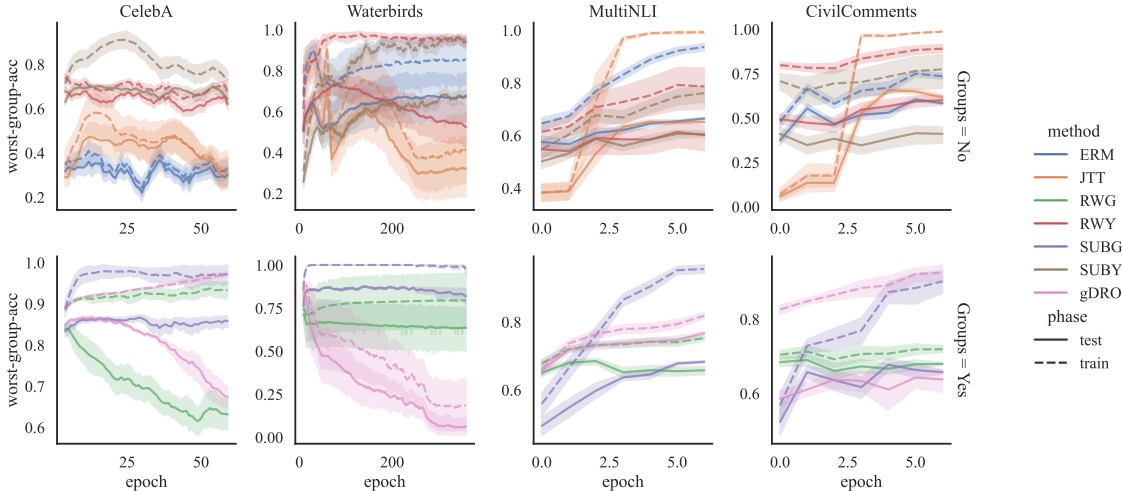

Figure 2: Average evolution of worst-group-accuracy for the top-5 best runs of each dataset and method. While reweighting methods (RWY, RWG, gDRO, JTT, ERM) sometimes degrade in performance over long training sessions, subsampling methods (SUBY, SUBG) show a more robust behavior.

(RWG and RWY) and gDRO, both degrade when removing regularization. That is consistent with findings of Słowik and Bottou (2021) where they establish close theoretical connections between gDRO and reweighting mechanisms. On the other hand, the subsampling method (SUBG) maintains its performance in the long run without the need of regularization. Byrd and Lipton (2019); Sagawa et al. (2019) reach a similar conclusion: strong regularization is necessary to benefit from data reweighting. Table 4 ratifies this, since SUBG prefers smaller weight decays than RWG. The superior performance of SUBG suggests two conclusions. On the one hand, we favor subsampling under tight computational budgets, since the resulting models are faster to train and depend less on regularization hyper-parameters. On the other hand, the considered benchmarks seem solvable with small data, showing that *either the tasks at hand are too easy or that the reweighting methods fail to make good use of all the data*.

To conclude, we comment on one similarity between early-stopped reweighting and subsampling. Reweighting uses a weighted random sampler to produce minibatches containing an equal amount of minority and majority examples (in expectation). This weighted random sampler is with replacement due to the scarcity of minority examples. Therefore, for a small amount of epochs, the model has likely seen all the minority examples while only observed a *subsample* of the majority examples. More specifically, the number of observed unique majority examples after sampling $k$ times is on average $N_{\mathrm{maj}}(1 - (1 - \frac{1}{N_{\mathrm{maj}}})^k)$, where $N_{\mathrm{maj}}$ is the number of majority examples contained in the dataset. Given that the best RWG model stops after 3 epochs for CelebA, it observes only 44% of majority examples on average, which amounts to subsampling the majority group.

## 6. Conclusion

We have shown that simple data balancing baselines achieve state-of-the-art performance in four popular worst-group-accuracy benchmarks. While balancing groups leads to best worst-group-accuracy, balancing class labels obtains competitive performance even in the absence of attribute information. We have also revisited the critical importance of having access to attribute information in the validation set, necessary to perform model selection based on worst-group-accuracy. Therefore, hyper-parameter tuning for domain generalization under weak supervision remains an open problem (Gulrajani and Lopez-Paz, 2020). We have illustrated some differences between data reweighting and data subsampling, advocating to try data subsampling first, since (i) it is faster to train and thus allows more hyper-parameter exploration, (ii) has less reliance on regularization, and (iii) has a more stable performance during long training sessions. All in all, our results raise two questions. First, are our current worst-group-accuracy benchmarks expressing a real problem? If so, is there room to outperform simple data balancing baselines in these datasets?

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

## Appendix A. Supplementary material

| Dataset | Groups | Method | Hyperparameters | | | | Worst Acc |
|---|---|---|---|---|---|---|---|
| | | | $\log_{10}$(LR) | $\log_{10}$(WD) | Epoch | Batch Size | |
| CelebA | No | ERM | -4.0 | -1.0 | 39.8 | 128.0 | 79.7±3.7 |
| | | JTT | -3.0 | -2.0 | 31.0 | 32.0 | 75.6±7.7 |
| | | RWY | -4.0 | -2.0 | 18.8 | 2.0 | 82.9±2.2 |
| | | SUBY | -3.0 | -2.0 | 41.4 | 128.0 | 79.9±3.3 |
| | Yes | RWG | -5.0 | -1.0 | 6.2 | 32.0 | 84.3±1.8 |
| | | SUBG | -4.0 | -1.0 | 8.2 | 8.0 | 85.6±2.3 |
| | | gDRO | -5.0 | -4.0 | 15.4 | 64.0 | 86.9±1.1 |
| CivilComments | No | ERM | -4.0 | -4.0 | 2.8 | 4.0 | 61.3±2.0 |
| | | JTT | -5.0 | -2.0 | 4.2 | 32.0 | 67.8±1.6 |
| | | RWY | -3.0 | -4.0 | 4.2 | 32.0 | 67.5±0.6 |
| | | SUBY | -3.0 | -3.0 | 3.8 | 16.0 | 51.2±3.0 |
| | Yes | RWG | -5.0 | -3.0 | 3.0 | 4.0 | 72.0±1.9 |
| | | SUBG | -4.0 | -4.0 | 3.2 | 8.0 | 71.8±1.4 |
| | | gDRO | -3.0 | -3.0 | 4.2 | 32.0 | 69.9±1.2 |
| MultiNLI | No | ERM | -4.0 | -4.0 | 4.6 | 2.0 | 67.6±1.2 |
| | | JTT | -5.0 | -3.0 | 5.0 | 4.0 | 67.5±1.9 |
| | | RWY | -3.0 | -4.0 | 4.6 | 16.0 | 68.0±1.9 |
| | | SUBY | -4.0 | -3.0 | 4.4 | 4.0 | 64.9±1.4 |
| | Yes | RWG | -5.0 | -3.0 | 2.0 | 4.0 | 69.6±1.0 |
| | | SUBG | -4.0 | -3.0 | 5.6 | 2.0 | 68.9±0.8 |
| | | gDRO | -4.0 | -3.0 | 5.4 | 8.0 | 78.0±0.7 |
| Waterbirds | No | ERM | -4.0 | -3.0 | 257.4 | 4.0 | 85.5±1.0 |
| | | JTT | -3.0 | -4.0 | 289.4 | 4.0 | 85.6±0.2 |
| | | RWY | -5.0 | -1.0 | 109.4 | 4.0 | 86.1±0.7 |
| | | SUBY | -5.0 | -2.0 | 319.8 | 2.0 | 82.4±1.7 |
| | Yes | RWG | -5.0 | 0.0 | 3.0 | 2.0 | 87.6±1.6 |
| | | SUBG | -4.0 | -2.0 | 175.2 | 4.0 | 89.1±1.1 |
| | | gDRO | -5.0 | 0.0 | 6.0 | 4.0 | 87.1±3.4 |

Table 5: Best hyperparameters for each dataset and method. The last column shows the worst test accuracy corresponding to that run. The top runs are selected using worst validation accuracy. The blue color highlights smaller values while yellow highlights higher values.

| Dataset | Method | Label | Prediction | Sentence |
|---------|--------|-------|------------|----------|
| MultiNLI | gDRO | 0 | 0 | [CLS] with this latter idea in my mind , i examined all the coffee ##cup ##s most carefully , remembering that it was mrs . cavendish who had brought made ##mo ##ise ##lle cynthia her coffee the night before . [SEP] i examined the coffee cups , but forgot it was cynthia who prepared the coffee . [SEP] |
| MultiNLI | RWG | 0 | 2 | [CLS] with this latter idea in my mind , i examined all the coffee ##cup ##s most carefully , remembering that it was mrs . cavendish who had brought made ##mo ##ise ##lle cynthia her coffee the night before . [SEP] i examined the coffee cups , but forgot it was cynthia who prepared the coffee . [SEP] |
| MultiNLI | SUBG | 0 | 2 | [CLS] with this latter idea in my mind , i examined all the coffee ##cup ##s most carefully , remembering that it was mrs . cavendish who had brought made ##mo ##ise ##lle cynthia her coffee the night before . [SEP] i examined the coffee cups , but forgot it was cynthia who prepared the coffee . [SEP] |

Figure 3: Cherry-picked sentence evaluated using the three methods "gDRO" "RWG" and "SUBG". The highlight in green at position $i$ shows the norm of the gradient the contradiction's class probability with respect to the input embedding at position $i$. i.e. $\|\nabla_{e_i}\hat{p}(\text{label} = \text{contradiction}|x)\|$, where $\hat{p}$ is the model's output probability given a sentence $x$. It is a proxy to the importance of each word in the final decision of the model as to whether it is a contradiction or not. Group DRO picked up on the crucial word "remembering" in the first sentence and therefore could correctly classify the relation between the two sentences as a contradiction. The other methods lack this "feature" and therefore fail. This sentence serves only an illustrative purpose.

