# OpenReview forum: "Simple data balancing achieves competitive worst-group-accuracy"
_cclear.cc/CLeaR/2022/Conference — CLeaR 2022 Poster_

### Official Review · Reviewer_rWjE · 2021-11-23

**Confidence:** 4
**Overall Score:** 5

**Main Review:**

The paper presents some thought-provoking results. If the goal is to do well on worst-group accuracy, then simple reweighting techniques can do as well on benchmark datasets.The set of datasets is comprehensive  and proposed techniques are intuitive enough. I like the direction of this work.

However, some of the results are underwhelming, and I feel that the paper can be improved by more theory/conjecture/analysis that explains the results.
1. In table 2, gdro is better on multi-nli. Why? What is it about this dataset that makes reweighting less effective? Same for suby on civilcomments. What can we learn from these exceptions?

2. Given that there are exceptions, should we still use the complicated methods? Under what conditions would reweighting be ideal? This can be answered by either extending the simulated toy dataset or by theoretical analysis on a simple setup.

Overall, I'm unclear what to take away from this paper. Is the proposal that we should adopt reweighing techniques? or that the datasets needs to be more complicated? If the former, you need to explain the exceptions in Table 2 better. If the latter, it will be good to propose a dataset where you clearly see the difference between sota methods and reweighting (even a toy data is fine).


**Summary:**

Empirical results showing that simple reweighting techniques can perform comparably to sota methods for worst-group accuracy.

---

> ### Author Response · Authors · 2021-12-03
> **Some further investigations and explanations following remarks by reviewer 3**
>
> We are happy to hear that you have found our results “thought-provoking” and “intuitive”. Thank you for your remark on the superior performance of G-DRO in MultiNLI. It inspired some further investigation that we will add to the manuscript. We hope this helps clarify!
>
> **Question: “In table 2, gdro is better on multi-nli. Why? What is it about this dataset that makes reweighting less effective? Same for suby on civilcomments. What can we learn from these exceptions?”**
>
> We believe gDRO is performing best in Multi-NLI because of the nature of the dataset. The spurious attribute “presence of: ‘no’, ‘never’, ‘nobody’ or ‘nothing’ in the second sentence” only helps in a small proportion of the data (see table 1). It is therefore not a dominating spurious correlation and the classifier still needs to extract other features to achieve good accuracy. Group dro minimizes a soft maximum of the group losses, thus enabling a more flexible way to penalize this non-dominant spurious correlation.
>
> Therefore, the simpler baselines are less effective in this case because balancing either through subsampling or reweighting is a strong measure that is meant to completely decorrelate the spurious feature with the label, which is most useful when the spurious correlation dominates the data. Therefore, the gains of getting rid of such a spurious correlation are compensated by the harsh capacity control imposed by balancing. Indeed, subsampling throws away a big proportion of the data, and reweighting has a similar effect when stopped early, which is the case for MultiNLI (training for only a few epochs).
>
> In the table shown in this link ([click here](https://i.imgur.com/Zd7HtRt.png)) we present a particular example that we found upon further investigation that serves to illustrate the hypothesis above; it shows a sentence evaluated using the three methods “gDRO” “RWG” and “SUBG”. The highlight in green shows the gradient norm of the contradiction’s class probability with respect to the input embeddings. It is a proxy to the importance of each word in the final decision of the model as to whether it is a contradiction or not. Group DRO picked up on the crucial word “remembering” in the first sentence and therefore could correctly classify the relation between the two sentences as a contradiction. The other methods lack this “feature” and therefore fail.
>
> As for SUBY, it seems to be underperforming for most datasets, even performing worse than ERM. We, therefore, argue that this would disqualify it from being one of the baselines that compete with more complicated methods.
>
> From these exceptions, we can learn that these simple balancing baselines might only work on simpler cases where the spurious correlation is clear cut and is present in a big majority of the data. In more nuanced cases such as MultiNLI, more complex methods such as gDRO might be useful. We will add this discussion in the paper and hope it clarifies things!
>
> **Question: “Given that there are exceptions, should we still use the complicated methods? Under what conditions would reweighting be ideal? This can be answered by either extending the simulated toy dataset or by theoretical analysis on a simple setup. Overall, I'm unclear what to take away from this paper. Is the proposal that we should adopt reweighing techniques? or that the datasets needs to be more complicated?”**
>
> Despite the existence of exceptions, simple data balancing baselines achieve state-of-the-art performance on average. In the end, our results show quantitatively that simple baselines are worth trying first. Furthermore, they give a very good ballpark of the performance that we should expect even from more complicated methods.
> The ideal conditions for reweighting (in contrast to subsampling) is when different groups are more similar, since reweighting uses more data. An extreme case would be when majority and minority groups are the same, and subsampling would just needlessly throw away i.i.d data.
>
> Our main takeaway is: **considering simple data balancing strategies is useful and important when optimizing for worst-group-accuracy, as they achieve competitive performance across all commonly studied benchmarks.**
>
> We do actually believe that more complicated datasets would be necessary to illustrate the strengths and weaknesses of different worst-group-accuracy optimizers. The fact that we can achieve such good performance by throwing away a lot of data as subsampling does is a clear pointer towards the fact that these datasets are very simple. In fact, we believe creating more realistic benchmarks is the way forward to improving our understanding about out-of-distribution generalization right now.

---

> > ### Comment · Reviewer_rWjE · 2022-01-02
> > **thank you**
> >
> > The analysis in response to the first question is very useful. will be great to add such interpretation to the paper.

---

### Official Review · Reviewer_PzXS · 2021-11-23

**Confidence:** 5
**Overall Score:** 8

**Main Review:**

Quality: The paper studies the problem that often arises when the data has groups/attributes that can be in the minority and often times get sacrificed at test time as the model can learn to rely on spurious features that work well for majority group at train time. The problem has gotten quite some attention after the work of gDRO from Sagawa et al. (2019) and IRM Arjovsky et al. (2019).
The paper carries out a careful empirical comparison of natural baselines that can be divided into two categories one that access group information and others do not access group information to recently proposed JTT and gDRO.
I found the results of the paper quite insightful, which I summarize below.
1. If we do not have access to attribute information, then rebalancing using class label is the go to method and not JTT.
2. If we have access to attribute information, then rebalancing using group label is approximately as good as gDRO barring some cases. Given its other benefits, it should still be off the shelf method to use in these settings.
3. Very interestingly, in the hardest problem when attribute information is also not used for validation then the benefit of using these simpler baselines is even more pronounced.
4. Also, in the hardest problem it seems when there is no attribute information (at train or validation time) and we are comparing RWY, SUBY, ERM, and JTT the winner seems to be SUBY.   Hence, the conclusion I draw is that if there is class imbalance, then SUBY is winner for worst group accuracy. So if there was no class imbalance, then SUBY may be still similar to ERM?
5. I would like to ask authors if they have intuitions or comparisons for the setting when there is no attribute information at train time, then methods like https://openreview.net/pdf?id=YygA0yppTR or https://openreview.net/pdf?id=b9PoimzZFJ seem to have shown impressive results. I ask this because authors conclude that it seems it's quite hard to beat these simple baselines.
6. Also, the authors at the end ask if the idea of worst group accuracy a good one. Does it not fundamentally boil down to asking if there is one classifier that can achieve a reasonably good performance (not far from their individually optimal ones) for all the groups? I believe in water birds and celebA this should be the case but I am not sure about civil environments and multiNLI dataset. A proxy for checking this can be that train a classifier separately on each group and find the individual optimal performance. Find how the classifier performs when transferred to other groups, if the gap is large then perhaps there is no common classifier that performs well across the groups. If the gap is small, then there exists a good classifier and thus worst group accuracy is meaningful. I wonder if authors have similar intuition?

Clarity: The paper is very clear.

Originality: The paper's contributions are new. The methods are natural and important baselines.

Significance: The paper is quite important.




**Summary:**

An insightful paper comparing important baselines to recently proposed approaches such as JTT and DRO

---

> ### Author Response · Authors · 2021-12-03
> **Additional clarifications and answers to reviewer's questions**
>
> We thank the reviewer for their thorough reading of the paper, the detailed feedback, and the interesting related work that they pointed out. We also thank the reviewer for his acknowledgment of our *careful* empirical evaluation, our *insightful* results, and our *comprehensive* choice of datasets.
>
> Below, we do our best to address the major concerns raised by the reviewer
>
> **Assertion: “... when there is no attribute information (at train or validation time) and we are comparing RWY, SUBY, ERM, and JTT the winner seems to be SUBY.”**
>
> We want to clarify that Table 3 shows that RWY is the least impacted by the absence of attribute information instead of SUBY. However, our main conclusion is that the presence of attribute information during validation is crucial for all methods since they all heavily suffer from its absence, even though RWY suffers less from it on average.
>
> **Question: “Hence, the conclusion I draw is that if there is class imbalance, then SUBY is winner for worst group accuracy. So if there was no class imbalance, then SUBY may be still similar to ERM?”**
>
> In the case of class balance, SUBY indeed becomes equivalent to ERM. RWY is also very similar to ERM in the case of class balance, except that samples are drawn with replacement in RWY.
>
> **Related work**
>
> We would like to thank the Reviewer for pointing out two related works.  [Diffenderfer et al. 2021] studies different forms of corruption on CIFAR and hence is inherently different from the specific task we attack in our submission. [Ahmed et al. 2021], however, is closely related to our work as it is concerned with inferring minority and majority groups and enforcing invariant solutions on them, which could be directly applied to our setting. Given its impressive performance on Coloured MNIST, we are keen to conduct similar experiments in non-synthetic setups such as CelebA, MultiNLI, or CivilComments in future work.
>
> **Assertion: “Also, the authors at the end ask if the idea of worst group accuracy a good one”**
>
> We are afraid that is not entirely true. The question we pose is whether the *current benchmarks* used in the worst group accuracy setting are enough to train models that can correctly discard the spurious correlations present in them. For instance, when a model is trained on a dataset with only red cars, we can hardly expect it to discard the color. Similarly, we wonder if the datasets here are enough to train models that would perform well on different stratifications of the same data. We will be looking for methods that disprove this by significantly outperforming our simple baselines, by starting with the two related works mentioned in the review.
>
> **Question: “Does it not fundamentally boil down to asking if there is one classifier that can achieve a reasonably good performance (not far from their individually optimal ones) for all the groups?”**
>
> We agree.
>
> **Question: “ A proxy for checking this can be that train a classifier separately on each group and find the individual optimal performance. Find how the classifier performs when transferred to other groups, if the gap is large then perhaps there is no common classifier that performs well across the groups. If the gap is small, then there exists a good classifier and thus worst group accuracy is meaningful. I wonder if authors have similar intuition?**
>
> We are afraid that this might not work in our precise setting as we stratify using the class values. The reason is that if we train a classifier in each group, the best one would be the constant classifier, always predicting the label associated with that group.

---

### Official Review · Reviewer_DmFu · 2021-11-24

**Confidence:** 4
**Overall Score:** 3

**Main Review:**

This paper conducts some experiments on the OOD generalization classification tasks. The authors test some sample reweighting methods on four benchmarks and find that simple data balancing baselines can achieve competitive performance. However, the technical contributions of this work are limited since it does not propose any new methods or theoretical analysis, for which I think this paper does not meet with the bar of this conference.


**Summary:**

This paper studies the Out-of-Distribution Generalization classification problem. And it test several simple reweighting methods on several benchmarks.

---

> ### Author Response · Authors · 2021-12-03
> **Official rebuttal**
>
> We thank the reviewer for their comment and acknowledge their point of view.
>
> The purpose of this submission is not to propose new methods or theory, but to take a sobering look at the rapidly growing field of worst-group-accuracy optimization, and show that very simple baselines do address all the benchmarks commonly considered in our community.
>
> In particular, recent years have provided us with a plethora of increasingly complex algorithms to solve the worst-group-accuracy benchmarks, relying on large computational budgets, multiple hyper-parameters, and blurring our understanding of why and when generalization happens. Our submission shows that incredibly simple baselines, requiring far less compute and no hyper-parameters, achieve state-of-the-art performance across most popular datasets. This is an important and novel result, calling for a closer look at both algorithms and benchmarks.
>
> In light of our results, new researchers in the field of worst-group-accuracy optimization have a solid reason to step back, implement and compare against simple subsampling baselines before endeavoring to develop novel, complex methods.

---

### Decision · Program_Chairs · 2022-01-12

**Decision:**

Accept (Poster)

**Comment:**

While it lacks theory/methods contributions, the paper provides a valuable and insightful empirical study of out-of-distribution generalization. At the same time, there are still remaining questions that arise from the authors' study that require exploration and explanation, which the reviewers have dutifully brought up. The authors provide some partial answers in the response. The authors should revise their paper to incorporate and build upon this discussion to make the study more complete, as would be expected from empirically-driven paper that focuses on existing methods.